# Effects of Visual–Motor Illusion via Image Videos Showing Increased Exercise Intensity on the Tibial Anterior during Sit-to-Stand Movement: A Study of Healthy Participants

Junpei Tanabe [1],*, Kazu Amimoto [2], Katsuya Sakai [3] and Motoyoshi Morishita [4]

[1] Department of Physical Therapy, Hiroshima Cosmopolitan University, Hiroshima 731-3166, Japan
[2] Department of Physical Therapy, Faculty of Rehabilitation, Sendai Seiyo Gakuin College, Miyagi 982-0114, Japan; kazamimoto125@yahoo.co.jp
[3] Department of Physical Therapy, Tokyo Metropolitan University, Tokyo 116-8551, Japan; sakai.katsuya@gmail.com
[4] Department of Physical Therapy, Reiwa Health Sciences University, Hukuoka 811-0213, Japan; m.morishita@rhs-u.ac.jp
* Correspondence: mjsstfive@gmail.com; Tel.: +81-82-849-6883

**Abstract:** Visual–motor illusion (VMI) elicits kinesthetic sensation from visual stimulation. We have previously performed ankle motion VMI with resistance applied to the ankle joint on the paralyzed side (power-VMI (P-VMI)) and ankle motion VMI without resistance (standard-VMI (S-VMI)) to activate the tibialis anterior (TA) muscle in stroke-paralyzed patients and compared sit-to-stand (STS) durations, but these studies did not measure TA activity during the STS movement. The purpose of this study was to evaluate the effects of different intensities of visual stimuli presented during VMI on TA and STS movement. Healthy right-footed adults ($n = 18$) observed two different VMI videos of ankle dorsiflexion, including S-VMI and P-VMI, with an observation time of 2 min each. STS movement was evaluated before and after watching each video. Each participant performed both S-VMI and P-VMI interventions on the same day. Only P-VMI enhanced the integrated electromyogram of the TA, increased the angular velocities of the trunk forward inclination and the ankle dorsiflexion, and shortened the STS duration. Our results indicate that P-VMI facilitates the activation of TA during STS, and we believe that we have clarified the intervention mechanism of VMI.

**Keywords:** visual–motor illusion; motor imagery; sit-to-stand; tibialis anterior muscle; sense of agency

## 1. Introduction

A reduced sit-to-stand (STS) ability is a very significant limiting factor in daily life activities [1]. In the STS movement, when the buttocks are brought off the seat, the floor reaction forces on the lower extremities are greater than those in a resting standing position [2]. Thus, performing this task is difficult for older adults with reduced muscle outputs and for clinical populations such as stroke hemiplegic patients (hemiplegic patients). The most active muscle when the buttocks are brought off the seat is the tibialis anterior [3]. In healthy individuals, the activity of the tibialis anterior muscle facilitates forward trunk inclination and ankle dorsiflexion during the STS movement, thereby affecting the forward horizontal velocity of the center of gravity (COG) [4]. Contrarily, previous studies on STS abilities in hemiplegic patients have reported decreased tibialis anterior muscle activity during the STS action [5,6]. Consequently, in hemiplegic patients, the ankle dorsiflexion angle is decreased when the trunk and shank are tilted forward to bring the buttocks off the seat, resulting in the prevention of the COG from advancing forward and prolonging the duration of the STS movement [7]. Therefore, the tibialis anterior muscle has been shown to influence the smoothness of the STS movement in healthy subjects and hemiplegic patients [3,4,8].

Recent research has demonstrated that visual–motor illusion (VMI) can elicit kinesthetic sensations through visual stimulation [9–11]. VMI employs a video image displayed on a monitor to mimic limb motion, creating the illusion that the patient's limbs are in motion, even though they are not physically moving [12–17]. Similarly, as another intervention that uses illusions, the effectiveness of mirror therapy has been reported [18]. Mirror therapy's distinctive feature is inducing a kinesthetic sensation in a paretic hand or leg through the observation of mirrored movements from a non-paretic hand or leg. However, mirror therapy may promote interhemispheric inhibition because it shows the movement of the non-paretic side [13]. Additionally, it is easier for the observer to attend to the intervening limbs in VMI because the observer does not have to move the other hand [13]. VMI in healthy participants increases the excitability of corticospinal tracts and induces the same motor imagery and brain activity as when the action in the observed video is actually performed [11,13]. Neural activity across multiple brain regions, including increased excitability of the frontoparietal network, has been associated with VMI [9–11]. Furthermore, VMI may activate brain regions (i.e., prefrontal area) associated with a sense of agency (SOA) [19]. Action observation therapy (AOT), in which motor imagery is induced from visual stimuli, has also been reported [20,21]. AOT activates the mirror neuron system of the observer's brain to mimic the observed person's movements [20,21]. The frontal-parietal network activated by VMI overlaps with the mirror neuron system activated by AOT [13]. VMI and AOT have been compared in brain activity studies of VMI. Sakai et al. [12] conducted VMI and AOT of finger flexion and extension in the right upper limb of healthy subjects and examined the resting-state functional connectivity (RSFC). Results showed that the RSFC in the premotor and parietal cortex increased after VMI compared to AOT. Therefore, VMI activated the frontal-parietal network contralateral to the upper limb that received the VMI. These brain areas are associated with kinesthetic illusion, sense of body ownership, and motor execution [12].

In our recent study, we found that applying VMI to the ankle joint on the paralyzed side leads to an increase in the automatic ankle dorsiflexion angle and reduces asymmetry during the STS movement in hemiplegic patients [22]. VMI is hypothesized to enhance both ankle dorsiflexion and STS capability by selectively engaging the corticospinal tract responsible for tibialis anterior muscle contraction during ankle joint VMI in healthy participants, as indicated in a previous study [10], and the activation of frontal-parietal motor-related connectivity in the brain [11]. However, VMI does not seem to reduce the STS duration [22]. Previous VMI intervention studies used video images that simply replicated the ankle dorsiflexion movement (i.e., in the standard (S-VMI)) [10,22]. The intensity of the ankle dorsiflexion exercises in the video images might have been too weak to stimulate the activation of the tibialis anterior muscle, which is necessary to reduce STS duration. As noted, VMI elicits a similar motor imagery and brain activity as when the action in the video is actually performed [11,13], suggesting that brain activity changes depend on the joint movement presented by video images. Prior research has indicated that the engagement of the brain area of motor-related areas is higher during maximal-intensity toe flexion exercises compared to moderate-intensity toe flexion exercises in healthy participants [23]. Furthermore, the extent of the increase in corticospinal tract excitability, as evaluated through motor-evoked potentials (MEPs) using transcranial magnetic stimulation (TMS), is contingent upon the intensity of the imagined or perceived muscle contraction [24]. From a cognitive psychology perspective, the SOA exhibits a connection with corticospinal tract excitability [25], with the added dimension that imbuing a sense of effort serves to augment the SOA [26]. Due to its ability to evoke motor imagery akin to that invoked by actual performance of the observed actions, P-VMI (more power-based VMI), characterized by ankle joint dorsiflexion movements at maximum intensity, is likely to elicit a notable upsurge in brain activity and corticospinal tract excitability when contrasted with S-VMI. Therefore, P-VMI may sufficiently activate the tibialis anterior muscle to result in decreased STS durations compared to S-VMI, which has been used in previous studies.

We performed P-VMI and S-VMI on the paralyzed side of the ankle joint of hemiplegic patients and compared their effectiveness [27,28]. The results showed that P-VMI improved the dorsiflexion function of the paralyzed ankle joint more than S-VMI, resulting in a shorter STS duration. However, these studies did not measure TA activity during the STS movement, and it is unclear whether VMI actually stimulates TA activity during the STS movement [27]. In addition, to date, no studies have measured and examined muscle activity in the effect of VMI on STS movement.

The objective of this study was to confirm whether P-VMI, which involves joint movements with increased exercise intensity, is more effective than S-VMI in stimulating TA activity during STS and shortening the duration of STS in healthy adults, and to determine the mechanisms underlying VMI.

## 2. Materials and Methods

### 2.1. Participants

The study participants included 18 healthy adults (14 men, 4 women; age = 26.6 ± 4.1 years (mean ± SD)) who were judged to be right-footed using Chapman's dominant foot test [29]. The Chapman's dominant foot test has 13 evaluation items, including standing on one foot and kicking a ball, among others. If participants used their left foot, they scored 3 points; if they used their right foot, they scored 1 point; if they used both feet, they scored 2 points. The criterion for determining the dominant foot is a score within the range of 11–27 points for right-footed and ≥28 points for left-footed. The study objective was explained to the participants, and written informed consent from each participant was obtained in compliance with the Declaration of Helsinki of 1975, as revised in 2000. Moreover, all procedures were conducted in compliance with relevant laws and institutional guidelines and this study received approval from our hospital's Ethics Committee under the approval number 1801 and was duly registered in the University Hospital Medical Information Network Clinical Trials Registry (UMIN CTR number: UMIN000042431).

### 2.2. VMI Interventions

The imaging videos used to demonstrate VMI were ankle dorsiflexion movement under two different conditions. In the VMI intervention, the ankle dorsiflexion projection in the video was performed using the participant's ankle dorsiflexion movement. In the initial video, ankle dorsiflexion was depicted with a TheraBand (Thera-band, Abilities, Tokyo, Japan) wrapped around the foot but without any tension applied (S-VMI). In the second video, the TheraBand was employed to add resistance to the foot, enabling dorsiflexion to be performed with maximal effort (P-VMI). In other words, both image videos showed the TheraBand, but one was not under tension (S-VMI), whereas the other was (P-VMI). The motion of the ankle joint was recorded for the dominant foot. The ankle dorsiflexion frequency for both conditions was set at 1 Hz, and video images were captured by a small LCD screen (iPad Pro, Apple, Cupertino, CA, USA). In this study, interventions were performed on the non-dominant foot only to ensure that all participants received interventions under the same conditions. Before the intervention, the ankle dorsiflexion image videos of the dominant foot were inverted via the software and converted to ankle dorsiflexion image videos of the non-dominant foot. In addition, before the intervention, participants were tasked with distinguishing whether the foot depicted in the image video represented S-VMI or P-VMI. All participants accurately identified whether the foot depicted in the image video was S-VMI or P-VMI. The VMI intervention is shown in Figure 1. Participants were seated in chairs with a table placed over their left foot which was covered with a black cloth so that they could not directly see their non-dominant left foot. The tablet computer was placed on the left ankle joint, and the ankle joint in the image videos was placed in such a way that it overlapped with the left ankle joint. When participants observed the video, they were given the following instructions: "As you do not have to actually move, please imagine that you are performing your own ankle movement in the video image". The VMI intervention lasted 2 min.

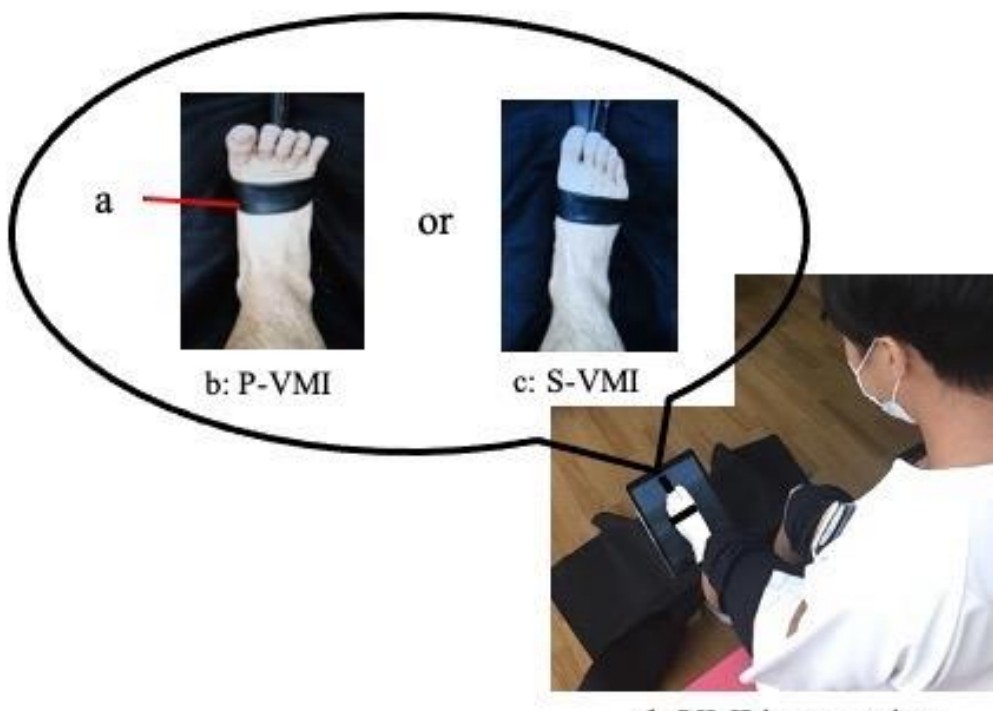

**Figure 1.** VMI intervention. a: TheraBand: resistance was applied to the ankle dorsiflexion movement using a TheraBand. b: The foot was subjected to resistance using the TheraBand for the purpose of executing maximum-effort dorsiflexion. c: S-VMI: A TheraBand was wrapped around the foot, but it remained without any applied tension. d: VMI intervention: participants viewed the video while seated for a duration of 2 min.

In P-VMI, a robust contraction of the tibialis anterior muscle, extension movement of the toes, and heightened activity in the tendons of the extensor hallucis longus muscle and extensor digitorum longus muscle were observed, clearly contrasting with S-VMI.

*2.3. Study Protocol*

Each participant performed S-VMI and P-VMI interventions on the same day. The control intervention was defined as a 2 min resting position without VMI. The P-VMI, S-VMI, and control interventions were defined as the P-condition, S-condition, and C-condition, respectively. The evaluations were conducted in a blinded assessor fashion. The order of the tasks was as follows: all participants started with C-conditions, followed by P- (Sequence 1) or S-conditions (Sequence 2). The time between the S-condition and P-condition was 2 min. The study protocol is summarized in Figure 2. Participants were sequentially allocated to Sequence 1 or 2 [30]. The specific VMI intervention was performed by rearranging the sequence in which the two groups observed the videos to prevent order effects and reduce the effects of mental fatigue.

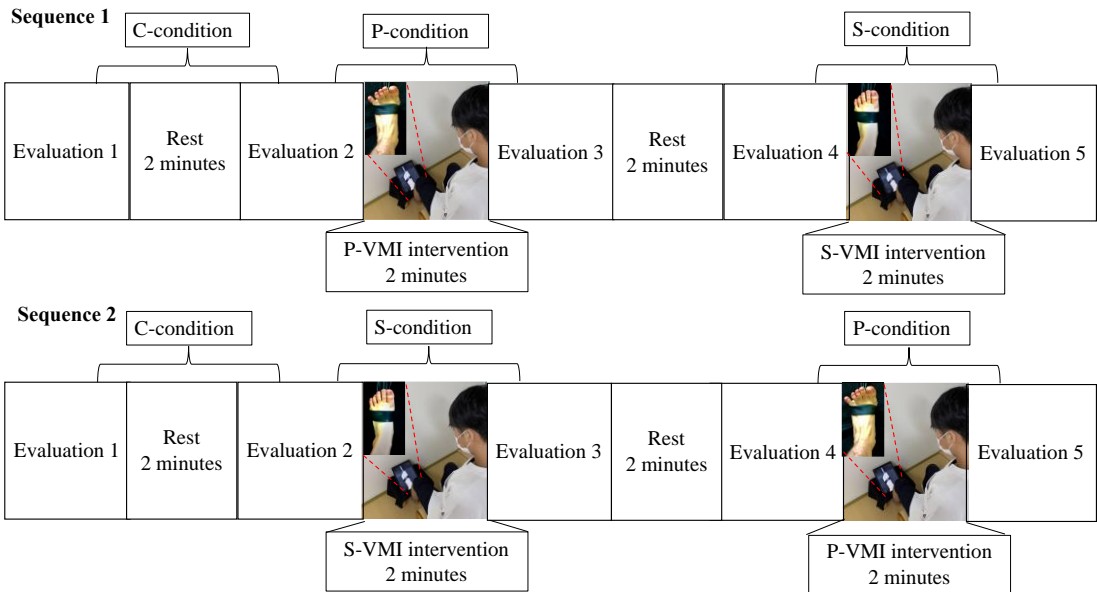

**Figure 2.** Schematic of the study protocol. Using Sequence 1 as an example, evaluation 1 was first performed as the C-condition, followed by 2 min of rest; evaluation 2 was then conducted. Thereafter, P-VMI was performed as the P-condition, and evaluation 3 was conducted. After the second rest period, evaluation 4 was conducted, the S-VMI was performed as the S-condition, and evaluation 5 was conducted. In Sequence 2, the order of the implementation of the P-condition and S-condition was switched. By putting the control condition first, we considered that it would be the value of the pure control condition, unaffected by P-VMI and S-VMI. In P-VMI and S-VMI, the order was swapped in Sequences 1 and 2 to eliminate the respective carryover effects.

### 2.4. Outcome Measurements

The primary outcome was the STS duration. Secondary outcomes included the trunk forward inclination angle, the trunk forward inclination angular velocity during the STS, the ankle dorsiflexion angle and ankle dorsiflexion angular velocity on the non-dominant side during STS, the active ankle dorsiflexion angle and active ankle dorsiflexion angular velocity on the non-dominant side, integrated electromyogram (iEMG) of the tibialis anterior muscle, and the degree of SOA during VMI.

### 2.5. Kinematic Measurements

To capture kinematic data of the trunk and ankle joint throughout the STS movement, we employed a video camera (EX-FC150, Casio, Tokyo, Japan), positioned on the non-dominant side of the participants to record the STS movement from the sagittal plane. The sampling frequency of the digital camera was 240 Hz. We affixed five markers to key anatomical landmarks, including the acromion process, greater trochanter, lateral epicondyle of the femur, lateral malleolus, and the 5th metatarsal head, on the non-dominant side of the participants [22,27,28]. These markers were strategically placed to facilitate precise movement measurements for subsequent analysis.

### 2.6. EMG Measurements

To collect iEMGs of the tibialis anterior muscle, we used a surface electromyogram (Bio Amp, FE-136, AD Instruments Pty, Ltd., New South Wales, Australia) and 44.8 mm × 22 mm oval disposable Ag/AgCl electrodes (Blue sensor N-00-S, Ambu, Baltorpbakken, Denmark). Electrode placements were based on SENIAM's (surface EMG for non-invasive assessment of muscles) European Recommendations for Surface Electromyography [31]. The electrodes for the tibialis anterior were positioned at a point approximately one-third along the line between the fibula head and the medial malleolus, aligned parallel to the muscle fibers. A ground electrode was attached to the fibula head.

### 2.7. Subjective SOA

The SOA induced during the VMI intervention was based on self-reports of participants on a visual analogue scale (VAS; ranging from 0 mm [no SOA] to 100 mm [an SOA occurs]). Participants were asked, "How much in control did you feel your own ankle joint movement?" [32]. The SOA was assessed in the P- and S-conditions.

### 2.8. Test Procedure

Performing STS from a low chair is considered a difficult task for healthy participants because of the increased load on the lower extremities [33]. In addition, Yoshioka et al. [34] defined a 40 cm-high chair as the normal height and reported that STS from a chair with a height lower than 40 cm required more activity of the tibialis anterior muscle [35]. In healthy subjects, STS from the 40 cm-high chair may have a lower task difficulty. Therefore, we postulated that a low chair (a 20 cm-high chair), which requires more tibialis anterior muscle activity, would be more challenging than the high chair (40 cm-high chair), which requires less activity of the tibialis anterior. Therefore, we hypothesized that the activation of the tibialis anterior muscle due to the influence of VMI would be more reflected during STS from a 20 cm-high chair. To ensure the consistency of the sitting posture, participants were asked to maintain a distance of 20 cm between both feet with both knee joints flexed at a 110° angle, arms folded in front of their chest, and pelvis tilted forward to prevent their trunk from tilting backward. Participants received verbal instructions to gaze straight ahead and to rise from their seated position promptly. To maintain consistency in buttock and foot placement across all trials, marks were delineated on both the seat and the ground. The STS movement was evaluated three times per evaluation [36]. Simultaneous electromyographic and kinematic measurements were carried out as participants executed the STS movement. The measurement of SOA was evaluated immediately after the STS evaluation. The time from VMI intervention to STS evaluation was approximately 30–60 s.

### 2.9. Kinematic Data Processing

A two-dimensional (2D) motion analysis system (ToMoCo-Lite, Toso System, Saitama, Japan) was employed to analyze the recorded videos, allowing for the computation of joint angles and the evaluation of joint movement time [36]. The reliability of angle calculations using Tomoco-Lite has been previously reported with intraclass correlation coefficients ranging from 0.80 to 0.97 [37], indicating a high intratester reliability [37]. The trunk forward inclination angle was defined as the angle between the line connecting the acromion process and the greater trochanter and a vertical line [22,27] (Figure 3). The ankle joint angle was determined as the angle formed by the line connecting the lateral epicondyle of the femur and the lateral malleolus, and the line connecting the lateral malleolus to the fifth metatarsal, as described in previous studies [22,27] (Figure 3).

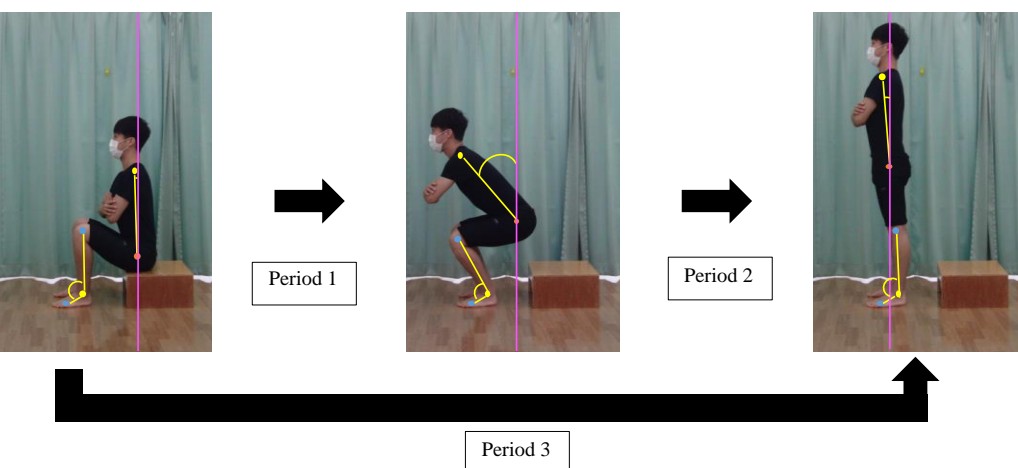

**Figure 3.** Period 1 was characterized as the interval from the initiation of the trunk forward inclination movement to the point of achieving the maximum ankle dorsiflexion angle on the non-dominant side. Period 2 was characterized as the interval from the attainment of the maximum ankle dorsiflexion angle on the non-dominant side to the achievement of a fully upright standing posture. Period 3 was characterized as the interval from the initiation of trunk forward inclination movement to the attainment of a fully upright standing posture.

### 2.10. STS Duration

The STS movement was defined in three distinct periods, labeled as periods 1, 2, and 3, in accordance with previous descriptions [38] (Figure 3). To determine the initial and final points of the STS movement, we computed the average angle and standard deviation based on ten consecutive frames capturing the unchanged trunk forward inclination angle immediately preceding the STS movement and following the STS movement utilizing a 2D motion analysis system. Following this, we determined the initiation and conclusion points of the STS movements by adding twice the standard deviation to the mean angle of the 10 consecutive frames capturing the unchanged trunk forward inclination angle, as outlined in a prior study [39]. We computed the duration of the STS movement by dividing the number of frames needed for the three STS periods by 240, which corresponds to the sampling frequency (i.e., the number of frames required during the STS period/240) [22,27].

### 2.11. Movement of the Trunk and Ankle Joint during STS

The movements of the trunk and ankle joint were analyzed using a 2D motion analysis system to calculate joint angles and evaluate joint movement duration, as described in [36]. The trunk forward inclination angle during the STS movement was computed from the initiation of the trunk forward inclination to the point at which the maximum forward inclination angles were reached. The angular velocity of trunk forward inclination was determined by dividing the trunk forward inclination angle by the time it took for the forward inclination to occur.

The ankle dorsiflexion angle on the non-dominant side during STS was calculated starting from the onset of ankle joint movement on the non-dominant side and extending to the point where the maximum dorsiflexion angle of the ankle joint on the non-dominant side was achieved. The ankle dorsiflexion angular velocity was determined by dividing the non-dominant ankle dorsiflexion angle by the time taken for ankle dorsiflexion to occur.

### 2.12. Electromyographic Data Processing

The data obtained by surface electromyography were transferred to a personal computer using PowerLab 4/26(AD Instruments Pty, Ltd.), and analyses were carried out in LabChart (AD Instruments Pty, Ltd.). The sampling rate was set at 1 kHz, and the bandpass filter was set to 10–500 Hz. In the analysis of the electromyography waveform, the mean amplitude was evaluated for a 5 s resting state (i.e., baseline) following full-wave

rectification, and the standard deviation was calculated. The muscle activity of the tibialis anterior muscle during STS and VMI intervention was defined as the time from when the amplitude of the baseline full-wave rectified waveform exceeded two standard deviations (2SDs) until the amplitude decreased by 2SDs [40]. Then, the muscle contraction time and iEMG were determined, and the iEMG per second was calculated. The value was normalized by dividing it by the 100% maximum voluntary contraction (MVC) of the tibialis anterior muscle (%). The 100% MVC of the tibialis anterior muscle was measured before the STS evaluation by manually applying resistance to the participants with ankle plantar dorsiflexion of $0°$ in a chair sitting position for 5 s [41]. The value of the 100% MVC was calculated via the iEMG, and 1 s with a stable amplitude was used [41].

*2.13. Statistical Analysis*

A two-way repeated-measures analysis of variance (ANOVA) was utilized for the statistical analysis of each measurement item, considering three conditions (C-, P-, and S-conditions) and two assessment times (pre-test and post-test) as the factors. The condition and assessment times were utilized to assess the main effect and interactions among the factors. When both a main effect and an interaction were observed, a simple main effects test was conducted using the Bonferroni's post hoc test. A paired *t*-test was used to compare the SOA of P-VMI and S-VMI. The effect size of the two-way ANOVA was determined using η2, and the effect size of the multiple comparison test was determined using Cohen's d to calculate the magnitude of change in pre- and post-VMI [42]. Effect sizes were interpreted according to Cohen's classification [43], with $η^2$ values being categorized as indicative of a small effect (0.01), a medium effect (0.06), or a large effect (0.14), and d values being indicative of a small effect (0.20), a medium effect (0.50), or a large effect (0.8). All statistical analyses were performed in SPSS ver. 20 (SPSS, IBM, Chicago, IL, USA).

*2.14. Sample Size*

In estimating the sample size, we determined Cohen's f through a two-way repeated-measures ANOVA of primary outcomes using G*Power 3.1.9.2 (Düsseldorf University, Düsseldorf, Germany) for each additional participant. Recruitment ceased when the statistical power $(1–β)$ exceeded 0.95. Cohen's f was computed from the partial $η^2$ using SPSS [40]. The results of the interim analysis confirmed that the effect size of the main effect of the two-way ANOVA in Period 1 of the STS duration was f = 0.55, with a power of 1.0. Given these positive outcomes, we chose to stop the study with 18 participants.

**3. Results**

*3.1. STS Duration*

There was no significant difference in the pre-test values among the three conditions. No interaction was confirmed, but the main effect was confirmed in the period 1 duration $(F (1,17) = 8.727; p = 0.009; η2 = 0.05)$. Post hoc test results showed that the post-test values were significantly reduced compared with the pre-test values in the P-condition $(t (17) = 3.154; p = 0.006; d = 0.689)$ (Table 1). There were no significant changes between the S- and C-conditions.

**Table 1.** Results of STS measurements.

| | C-Condition | | P-Condition | | S-Condition | |
|---|---|---|---|---|---|---|
| | **Pre** | **Post** | **Pre** | **Post** | **Pre** | **Post** |
| STS duration (s) | | | | | | |
| Period 1 [c] | 0.69 (0.10) | 0.70 (0.08) | 0.69 (0.09) | 0.65 (0.07) [a,†] | 0.69 (0.08) | 0.67 (0.09) |
| Period 2 [d] | 0.66 (0.36) | 0.66 (0.38) | 0.65 (0.37) | 0.63 (0.32) | 0.65 (0.36) | 0.64 (0.35) |
| Period 3 [e] | 1.35 (0.42) | 1.36 (0.43) | 1.34 (0.42) | 1.28 (0.36) | 1.34 (0.41) | 1.32 (0.41) |
| Trunk and ankle joint movement | | | | | | |
| Trunk forward inclination angle (°) | 44.2 (7.9) | 43.5 (7.5) | 42.9 (6.4) | 44.2 (6.9) | 44.4 (7.8) | 44.2 (7.7) |
| Trunk forward inclination angular velocity (°/s) | 93.6 (18.4) | 93.3 (21.1) | 92.1 (18.0) | 104.8 (22.2) [a,b,§] | 97.4 (19.2) | 99.6 (18.2) |
| Ankle dorsiflexion angle (°) | 28.0 (6.9) | 27.9 (6.6) | 28.2 (7.0) | 28.4 (6.9) | 27.5 (6.7) | 27.7 (5.8) |
| Ankle dorsiflexion angular velocity (°/s) | 72.5 (23.2) | 70.2 (20.6) | 73.6 (21.1) | 86.1 (27.8) [a,b,§] | 75.9 (25.7) | 78.4 (23.3) |
| Tibialis anterior muscle activity | | | | | | |
| IEMG of the tibialis anterior muscle (%) | 49.0 (12.3) | 46.6 (14.2) | 47.3 (14.4) | 53.4 (14.0) [a,b,§] | 46.4 (15.1) | 49.4 (14.3) |

Data are presented as means (standard deviation). [a]: There is a significant difference between the pre and post values. [b]: There is a significant difference between the post values for P-conditions and C-conditions. [c]: Interval from the initiation of the trunk forward inclination movement to the point of achieving the maximum ankle dorsiflexion angle on the non-dominant side. [d]: Interval from the attainment of the maximum ankle dorsiflexion angle on the non-dominant side to the achievement of a fully upright standing posture. [e]: Interval from the initiation of trunk forward inclination movement to the attainment of a fully upright standing posture. [†] Effect size is medium. [§] Effect size is large.

### 3.2. Trunk and Ankle Joint Movements during STS

The outcomes of trunk and ankle joint movements are presented in Table 1. No significant differences were observed in the pre-test values among the three conditions. No significant main effects or interactions were observed for the angles of trunk forward inclination and ankle dorsiflexion. The main effect was observed in the angular velocities of trunk forward inclination and ankle dorsiflexion (trunk forward inclination angular velocity: $F_{(1,17)} = 9.017$; $p = 0.008$; $\eta2 = 0.06$; ankle dorsiflexion angular velocity: $F_{(1,17)} = 5.329$; $p = 0.034$; $\eta2 = 0.04$) and their interaction (trunk forward inclination angular velocity: $F_{(2,34)} = 4.294$; $p = 0.022$; $\eta2 = 0.07$; ankle dorsiflexion angular velocity: ($F_{(2,34)} = 5.032$; $p = 0.043$; $\eta2 = 0.08$). Post hoc test results showed that the post-test values were significantly higher than those of the pre-test in the P-condition (trunk forward inclination angular velocity: $t_{(17)} = -3.664$, $p = 0.002$, $d = 0.871$; ankle dorsiflexion angular velocity: $t_{(17)} = -3.753$, $p = 0.002$, $d = 0.886$). In addition, the post-test value of the P-condition was significantly higher than that of the C-condition (trunk forward inclination angular velocity: $t_{(17)} = -2.665$; $p = 0.048$; ankle dorsiflexion angular velocity: $t_{(17)} = -4.660$; $p = 0.003$). There were no significant changes between the S- and C-conditions.

### 3.3. Tibialis Anterior Muscle Activity during STS and VMI Intervention

The results of tibialis anterior muscle activity during STS are shown in Table 1. There was no significant difference in the pre-test values of the three conditions. The main effect and interaction were confirmed (main effect: $F_{(1,17)} = 4.283$, $p = 0.044$, $\eta2 = 0.04$; interaction: $F_{(2,34)} = 5.373$; $p = 0.009$; $\eta2 = 0.11$). Post hoc test results showed that the post-test values were significantly higher than the pre-test values in the P-condition ($t_{(17)} = -3.413$, $p = 0.003$, $d = 0.806$). In addition, the post-test value of the P-condition was significantly higher than that of the C-condition ($t_{(17)} = -3.74$; $p = 0.006$). There were no significant changes between the S- and C-conditions.

The tibialis anterior muscle activity during VMI intervention was below two SDs of the mean amplitude at rest for both S-VMI and P-VMI, confirming that no lower limb movement occurred during each intervention.

*3.4. SOA*

The SOA during VMI was 69.5 mm (SD = 8.3) and 58.0 mm (SD = 12.6) for the P- and S-conditions, respectively, indicating that P-VMI had a significantly higher SOA than S-VMI (t (17) = −6.772, *p* < 0.001).

## 4. Discussion

The P-condition showed a significant benefit in STS duration, angular velocities of the forward inclination of the trunk and dorsiflexion of the ankle, and iEMG of the tibialis anterior during STS movement. In addition, the SOA of P-VMI was significantly higher than that of S-VMI. The alignment between motor intention (i.e., motor imagery) and visual feedback can engender the SOA and exert an impact on the excitability of the corticospinal tract [25]. Additionally, Minohara et al. [26] found that the introduction of a sense of effort heightened the SOA. Hence, P-VMI, characterized by a heightened exercise intensity, elicits a stronger sense of effort compared to S-VMI. This, in turn, leads to an increased SOA and is believed to enhance the excitability of the corticospinal tract, consequently augmenting the effect on ankle dorsiflexion and primarily activating the tibialis anterior muscle. Additionally, according to Mizuguchi et al. [24], the excitability of the corticospinal tract was contingent upon the intensity of the imagined muscle contraction. In P-VMI, the tibialis anterior muscle contraction was strongly imagined by increasing the exercise intensity compared to the S-VMI. The tibialis anterior muscle activity during STS has been reported to be involved in the motive force of trunk forward inclination and the forward-lower movement of the COG, as well as shank forward tilts [3]. Moreover, the forward inclination of the trunk and ankle dorsiflexion during the STS movement affect the speed of movement of the front COG [4]. Therefore, P-VMI activated greater tibialis anterior muscle activity during STS, improved the angular velocities of trunk forward inclination and ankle joint dorsiflexion, and shortened the STS duration from starting at the initiation of the forward trunk inclination movement and extending to the point of maximum ankle dorsiflexion on the non-dominant side. Based on the above, P-VMI may have promoted greater corticospinal tract excitability, indicating a possible benefit to motor performance.

On the other hand, S-VMI did not affect STS duration, trunk inclination, ankle dorsiflexion movements, or tibialis anterior muscle activity. We attribute this to the influence of the model in the presented video images. A previous study reported that the frontal-parietal network is activated during VMI, with the involvement of the mirror neuron system [13]. Therefore, there is a common mechanism between VMI and action observation therapy (AOT). Previous studies of AOT have reported that observing a model with higher motor skills than the observer has a greater influence on motor performance [44]. In addition, most previous studies have presented image video models with a higher motor performance than the observer and have reported activation of brain activity and improvement of physical function [20,45]. However, since S-VMI only involves the ankle joint dorsiflexion movement, the motor skill of the image video model was low among healthy subjects, and it may not have been reflected in the rise in healthy subjects. Therefore, we believe that S-VMI had little effect on the activity of the tibialis anterior muscle, which was hypothesized to be affected by S-VMI of the ankle joint. In addition, it did not affect trunk and ankle joint movements.

This study has some limitations. First, neuroimaging was not used to assess the hypothesized brain activity and excitability of the corticospinal tract. Second, the dominant foot was not evaluated. Third, only the sagittal plane was assessed, and the lateral movement of the trunk was not investigated. Fourth, we did not collect detailed characteristics of the participants, such as whether they played any sports, their eyesight, or degree of illusion susceptibility. Fifth, this study did not measure TA activity in actual hemiplegic patients; the results were obtained in healthy subjects. In the future, it is necessary to examine the muscle activity of the TA during STS in actual hemiplegic patients. In addition, future studies should examine the task of the control condition. To eliminate the factor of consciousness, the control condition should include a task in which attention is focused on

the non-dominant foot. For example, in a previous study [12], as a control condition for the VMI intervention, a tablet was placed in front of each participant's hand so that it did not overlap with the real hand, and the movements of the tablet were observed.

Overall, our findings demonstrated that P-VMI has a significant impact on the STS movement and suggest that P-VMI enhances tibialis anterior muscle activity while standing, thereby shortening the STS duration. As the VMI method is simple and reasonable, it is easy to use in clinical practice. Therefore, it may be applicable for a large number of patients. Our results indicate that P-VMI facilitates activation of the TA during STS, and we believe that we have clarified the intervention mechanism of VMI.

### 5. Conclusions

This study examined whether two types of VMI on the ankle joint affect TA activity or STS duration. The results showed that only P-VMI increased the activity of the TA during STS movement and reduced the STS duration. To date, no studies have measured and examined muscle activity regarding the effect of VMI on the STS movement. Although this study examined a topic that has been of little help in the therapeutic field, it may help to clarify the mechanisms of the effects of VMI on the STS movement.

**Author Contributions:** J.T.: methodology, investigation, validation, writing—original draft, writing—review and editing. K.A.: methodology, validation, writing—review and editing. M.M.: validation, review, and editing. K.S.: methodology, validation, writing—review and editing. All authors have read and agreed to the published version of the manuscript.

**Funding:** This work was supported by JSPS KAKENHI grant number 23K16564.

**Institutional Review Board Statement:** All procedures were conducted in compliance with relevant laws and institutional guidelines and the study was approved by the Ethics Committee of our hospital (approval number 1801).

**Informed Consent Statement:** The study purpose was explained to the participants, and written informed consent from each participant was obtained in compliance with the Declaration of Helsinki of 1975, as revised in 2000.

**Data Availability Statement:** The data cannot be made available to readers upon request as we, the authors, do not have ethical approval to share the data.

**Acknowledgments:** We express our gratitude to the rehabilitation staff at Kurashiki Rehabilitation for their invaluable cooperation in this study.

**Conflicts of Interest:** The authors declare no conflict of interest.

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
