# Peer review of "Effects of Visual–Motor Illusion via Image Videos Showing Increased Exercise Intensity on the Tibial Anterior during Sit-to-Stand Movement: A Study of Healthy Participants"

_2035-8377, doi:10.3390/neurolint15040081_

Round 1

Reviewer 1 Report

This is a very interesting design to study to show the efficacy of different VMI interventions on TA and STS movement. The applications are immense for the old and stroke population. Thus, the main weakness of the paper is unavailability of patient population. it is unclear whether the same significant results observed in the healthy group are replicated in patient population.

The figures are blurry and must be improved. 

Author Response

Responses to Reviewers

We thank the reviewers for reviewing our manuscript and sincerely appreciate their insightful comments, which have helped us significantly improve our manuscript. We have provided point-by-point responses to the reviewers’ comments and indicated the changes made in the revised manuscript. We have marked our corrections in red.

Reviewer 1’s comments:

  1. This is a very interesting design to study to show the efficacy of different VMI interventions on TA and STS movement. The applications are immense for the old and stroke population. Thus, the main weakness of the paper is unavailability of patient population. it is unclear whether the same significant results observed in the healthy group are replicated in patient population.

Response: Thanks for pointing that out. We have added the following sentences to the Limitation 

Fifth, this study did not measure TA activity in actual hemiplegic patients because the results were obtained in healthy subjects. In the future, it is necessary to examine the muscle activity of TA during STS in actual hemiplegic patients. (Page 11 from line 419 to line 421)”

  1. The figures are blurry and must be improved.

Response: Thanks for pointing that out. We have corrected the Figure.

Reviewer 2 Report

Dear Authors,

the article needs some revision and  I am reporting my criticisms step-by-step 

TITLE: your scope is  ‘to evaluate the effects of different intensities of visual stimuli presented during VMI on TA’. For this reason, I suggest replacing the term ankle joint function with tibial anterior.

Introduction

line 33: you stated that decreased STS  ability is the major limiting factor for daily life activities. I can agree that STS is important for the autonomy of the movement but it is not the major limiting factor. Moreover, I read the reference that you cited, but I did not find information confirming your statement. Please, change the statement (very important instead major)  and try to search for a coherent bibliography  (see for example: Differences in Rehabilitation Needs after Stroke: A Similarity Analysis on the ICF Core Set for Stroke- Int. J. Environ. Res. Public Health 2020, 17, 4291; doi:10.3390/ijerph17124291)

line 49-50: MVI and mirror therapy are quite different: mirror therapy uses the patient's body part, and VMI is performed with an anonymous body limb. Do you think that MVI is different from action observation therapy? why did you describe AOT in the section Discussion but not here?

line 129: specify if the video ankle dorsiflexion motion is performed in the subjects recruited in the study or not (for the same reason (AOT? MVI? MT?)

Protocol of study:

lines 163 -172: you did not explain the reason for the order of the tests: C-P-S. Moreover, I do not understand why the STS performance pre-test is repeated for any condition. 

Results:

Table 1 reports all STS pre-test measures that are different in C-P-S condition.  How do you explain these differences?

Discussion: 

line 405:  ‘ the dominant foot was not evaluated’  incongruence with line 116: ‘right-footed using Chapman’s dominant foot test….’’

is it the same work??? or have you mixed two works? 

lines 421- 425: I think that you have only reported another piece of evidence on a topic far too much investigated but scarcely useful in the therapeutical field. 

your article can be published after minor revision 

Author Response

Responses to Reviewers

We thank the reviewers for reviewing our manuscript and sincerely appreciate their insightful comments, which have helped us significantly improve our manuscript. We have provided point-by-point responses to the reviewers’ comments and indicated the changes made in the revised manuscript. We have marked our corrections in red.

Reviewer 2’s comments:

  1. TITLE: your scope is ‘to evaluate the effects of different intensities of visual stimuli presented during VMI on TA’. For this reason, I suggest replacing the term ankle joint function with tibial anterior.

 Response: Thank you for your advice. We have corrected the title to "Effects of Visual-Motor Illusion by Image Videos Showing Increased Exercise Intensity on Tibial Anterior During Sit-To-Stand Movement: A Study of Healthy Participants". (Page 1 from line 1 to line 4)

  1. Introduction

line 33: you stated that decreased STS ability is the major limiting factor for daily life activities. I can agree that STS is important for the autonomy of the movement but it is not the major limiting factor. Moreover, I read the reference that you cited, but I did not find information confirming your statement. Please, change the statement (very important instead major) and try to search for a coherent bibliography (see for example: Differences in Rehabilitation Needs after Stroke: A Similarity Analysis on the ICF Core Set for Stroke- Int. J. Environ. Res. Public Health 2020, 17, 4291; doi:10.3390/ijerph17124291)

 Response: Thank you for pointing this out. We have revised it to "very important" based on the reviewer 2's opinion. Also, thank you for presenting the paper. (Page 1 line 32) We have changed our references to "Differences in Rehabilitation Needs after Stroke: A Similarity Analysis on the ICF Core Set for Stroke- Int. J. Environ. Res. Public Health 2020, 17, 4291; doi:10.3390/ijerph17124291)".

We added " Perin, C., Bolis, M., Limonta, M., Meroni, R., Ostasiewicz, K., Cornaggia, C. M., Alouche, S. R., da Silva Matuti, G., Cerri, C. G., Piscitelli, D.; Differences in Rehabilitation Needs after Stroke: A Similarity Analysis on the ICF Core Set for Stroke. Int J En-viron Res Public Health. 2020, 17(12), 4291. DOI: 10.3390/ijerph17124291. " to the reference. (Page 12 from line 457 to line 459)

We have removed "Khemlani, M. M.; Carr, J. H.; Crosbie, W. J. Muscle synergies and joint linkages in sit-to-stand under two initial foot posi-tions. Clin. Biomech. (Bristol Avon) 1999, 14 (4), 236–246. DOI: 10.1016/s0268-0033(98)00072-2" from the reference.

  1. line 49-50: MVI and mirror therapy are quite different: mirror therapy uses the patient's body part, and VMI is performed with an anonymous body limb. Do you think that MVI is different from action observation therapy? why did you describe AOT in the section Discussion but not here?

Response: Thank you for pointing this out. In a previous study, Kaneko et al. reported that the neural basis of mirror therapy and VMI may overlap somewhat because the two approaches involve psychological sensations of body possession and kinematic perception in mirrors and artificial bodies in movies. (Kaneko et al., 2016)" In addition, it has been reported that mirror therapy promotes abnormal interhemispheric inhibition because it moves the upper extremity without paralysis (Aoyama et al., 2021).

In other papers on VMI, the authors describe the characteristics of VMI and Mirror therapy in the introduction and describe the advantages of VMI (Kaneko et al., 2016; Aoyama et al., 2021). Therefore, in this study, we described mirror therapy in the introduction in accordance with previous studies. (Page 2 from line 51 to line 62)

Also, the description of action observation therapy has been added to the Introduction.

We added "Action observation therapy (AOT), in which motor imagery is induced from visual stimuli, has also been reported [22,23]. AOT activates the mirror neuron system of the observer's brain to mimic the observed person's movements [22,23]. The frontal-parietal network activated by VMI overlaps with the Mirror neuron system activated by AOT [14]. VMI and AOT have been compared in brain activity studies of VMI. Sakai et al. [12] conducted VMI and AOT of finger flexion and extension in the right upper limb of healthy subjects and examined resting-state functional connectivity (RSFC). Results showed that RSFC in premotor and parietal cortex increased after the VMI compared to AOT. Therefore, VMI activated the frontal-parietal network contralateral to the upper limb that received the VMI. These brain areas are associated with kinesthetic illusion, sense of body ownership, and motor execution [12]. Furthermore, brain activity during motor imagery after VMI was associated with stronger connectivity in the right frontal-parietal network compared to AOT, resulting in clearer motor imagery [13]." (Page 2 from line63 to line75)

We have added the following two papers to REFERENCES

Ertelt, D., Small, S., Solodkin, A., Dettmers, C., McNamara, A., Binkofski, F., Buccino, G. Action observation has a positive impact on rehabilitation of motor deficits after stroke. Neuroimage. 2007, 36, 164-173. DOI: 10.1016/j.neuroimage.2007.03.043

Rizzolatti, G., Fadiga, L., Gallese, V., Fogassi, L. Premotor cortex and the recognition of motor actions. Brain Res Cogn Brain Res. 1996, 3(2), 131-141. DOI: 10.1016/0926-6410(95)00038-0

  1. line 129: specify if the video ankle dorsiflexion motion is performed in the subjects recruited in the study or not (for the same reason (AOT? MVI? MT?)

Response: Thanks for the advice. The dorsiflexion image of the ankle joint without video is the subject's own dorsiflexion image of the ankle joint. Therefore, we have added "In the VMI intervention, the ankle dorsiflexion projection in the video was performed using the participant’s ankle dorsiflexion movement."(Page 3 from line 131 to line 132)

  1. lines 163 -172: you did not explain the reason for the order of the tests: C-P-S. Moreover, I do not understand why the STS performance pre-test is repeated for any condition.

Response:  

(About the order of the tests)

By putting the control condition at the first, we considered that it would be the value of the pure control condition, unaffected by P-VMI and S-VMI.In P-VMI and S-VMI, the order was swapped in Sequences 1 and 2 to eliminate the respective carryover effects.

We have added to the text the reason for our order. (Page 5 from line 182 to Line 185)

(About repeating the pre-test.)

Since this study was conducted under three conditions on one participant, carryover effects were a concern. Therefore, we thought we could confirm the carryover effect by evaluating and comparing pre-test values.

  1. Results:

Table 1 reports all STS pre-test measures that are different in C-P-S condition.  How do you explain these differences?

Response: Three conditions were performed on one subject. Thus, we evaluated the pre-test values to check for changes due to carryover effects.

In fact, we compared the pre-test values of the three conditions and found no significant difference, suggesting that the carryover effect could be suppressed.

We have included in the results section the results of comparing the pre-test values of the three conditions. (Page 8 Line 325)

  1. Discussion:

line 405: ‘the dominant foot was not evaluated’ incongruence with line 116: ‘right-footed using Chapman’s dominant foot test….’

is it the same work??? or have you mixed two works?

Response: We took an ankle dorsiflexion image of the dominant foot and inverted it to create an ankle dorsiflexion image of the non-dominant foot (left foot). Therefore, the intervening foot was the non-dominant foot (left foot) and not the dominant foot.

Therefore, we have stated " Second, the dominant foot was not evaluated."

In the VMI intervention, we have listed "In this study, interventions were performed on the non-dominant foot only to ensure that all participants received interventions under the same conditions. Before the intervention, the ankle dorsiflexion image videos of the dominant foot were inverted via the software and converted to ankle dorsiflexion image videos of the non-dominant foot. In addition, before the intervention, participants were tasked with distinguishing whether the foot depicted in the image video represented S-VMI or P-VMI. All participants accurately identified whether the foot depicted in the image video was S-VMI or P-VMI. The VMI intervention is shown in Figure 1. Participants were seated in chairs with a table placed over their left foot and covered with a black cloth so that they could not directly see their non-dominant left foot." (From Page3 line140 to Page 4 line 149)

  1. lines 421- 425: I think that you have only reported another piece of evidence on a topic far too much investigated but scarcely useful in the therapeutical field.

Response: Thank you for pointing this out.We have corrected from " This study may help to clarify the mechanism of the effect of VMI on STS movement." to "Although this study examined a topic that has been little help in the therapeutic field, it may help to clarify the mechanisms of the effects of VMI on STS movement."(Page 11 from line 437 to line 439)
